# Forward-Looking Sonar-Based Stream Function Algorithm for Obstacle Avoidance in Autonomous Underwater Vehicles

Moon Hwan Kim [1], Teasuk Yoo [2], Seok Joon Park [2] and Kyungwon Oh [3],*

[1] School of Electrical and Electronic Engineering, Yonsei University, Seoul 03722, Republic of Korea; moonani.kim@yonsei.ac.kr
[2] LIG Nex1, Seongnam-si 13488, Republic of Korea; teasuk.yoo@lignex1.com (T.Y.); seokjoon.park2@lignex1.com (S.J.P.)
[3] Department of Aircraft MRO Engineering, Howon University, Gunsan-si 54058, Republic of Korea
* Correspondence: kwoh@howon.ac.kr

**Abstract:** Autonomous Underwater Vehicles (AUVs) have emerged as pivotal tools for intricate underwater missions, spanning seafloor exploration to meticulous inspection of subsea infrastructures such as pipelines and cables. Although terrestrial obstacle avoidance paradigms exhibit proficiency, their efficacy diminishes in aquatic environments due to the nuanced challenges and distinct dynamics inherent to marine realms and AUV maneuvering. This paper presents an advanced obstacle avoidance algorithm for AUVs based on a stream function framework. Central to this approach is the utilization of a stream function, further nuanced by a radial histogram that serves as the defining cost function. This work also encapsulates constraints related to the maximum allowed path curvature, ensuring enhanced path optimization. Comprehensive simulation results validate the robustness and adaptability of the introduced strategy, evincing its capacity to outline both practicable and optimal evasion trajectories across diverse operational contexts.

**Keywords:** stream function; path-planning algorithms; obstacle avoidance AUV; fluid dynamics in path planning

## 1. Introduction

AUVs are becoming increasingly important in underwater operations such as seafloor exploration and the inspection of pipelines or cables. Their use not only promises cost savings, but also reduces the risks associated with systems that require a human operator on board [1]. A major challenge in the development of advanced autonomous systems is to create real-time path planning and obstacle avoidance strategies that can effectively guide the vehicle through unstructured environments.

Research in recent times has seen the development of numerous strategies aimed at addressing the intricacies of path planning. Fundamentally, path-planning algorithms can be categorized into pregenerative and reactive types [2]. The former, often known as global path-planning algorithms, determines the path prior to the mission's commencement. Examples include the Free Space Network, which utilizes a directed graph reflecting environmental properties [3] and the Cell Decomposition approach, which employs an undirected graph based on subdividing the environment into distinct predefined cellular structures [4]. Another notable technique is the Octree-based method, optimal for three-dimensional environments, although it requires recursive subdivisions of mixed cells [5]. A notable limitation of these global algorithms is their rigidity in adapting paths during active missions due to their nonreal-time nature, although they excel in delivering collision-free paths.

However, reactive algorithms predominantly employ differential computational methods. The Potential Field method, for instance, traces the gradient of artificially produced

potential field lines in an environment. However, this could lead to local minimum challenges, which could ensnare the vehicle or produce suboptimal solutions [6]. Solutions such as the stream function, a variant of the harmony function, have been proposed to counteract this [7]. This technique has witnessed its applicability in areas such as robotic navigation, with instances of using hydrodynamic analysis to develop stream functions for intricate geometries [8]. In particular, most existing research is focused on terrestrial and aerial vehicles, leaving underwater applications relatively unexplored. The underwater environment presents unique challenges. Communication is hampered due to the restrictive bandwidth in the underwater channels. The domain is susceptible to currents and may span worldwide [9]. Additionally, torpedo-like vehicles exhibit strong non-holonomic characteristics, with path curvature constraints stemming from vehicle controllability and sonar system limitations. The limited literature focusing on the planning of underwater-specific routes accentuates the need for innovative algorithms tailored for these demanding conditions [1].

Recently, there has been a heightened interest in examining the path-planning capabilities of AUVs through research based on streamline principles. For example, Yongqiang et al. [10] verified the suitability of the stream function for formation control and path planning within multi-agent frameworks. This paradigm also extends to wireless sensor networks, where Wang et al. [11] advocated the application of the stream function to enhance network coverage. Deeper in marine dynamics, To et al. [12] advanced a local motion plan for underwater vehicles, employing the stream function to increase maneuverability. Furthermore, Nan et al. [12] embarked on a rigorous exploration of the path identification algorithm in conjunction with the stream function. W. Cai and et al. [13] propose a fluid mechanics-based obstacle avoidance method for AUVs in 3D IoUT, enhancing energy efficiency and multi-obstacle avoidance through path deformation and energy models. P. Yao et al. [14] propose an improved IIFDS-based submerged path-planning method for autonomous underwater vehicles in intricate ocean settings. The culmination of these studies underscores the pivotal role of streamline-based methodologies in optimizing the functionality of various automated systems. Kazimierski et al. [15] conducted an in-depth examination of process noise, commonly characterized as Gaussian noise with an author-defined covariance matrix. Their approach combined rigorous analytical methods with hands-on empirical testing, validating their theoretical insights against established benchmarks in underwater target tracking.

This paper introduces a two-dimensional path-planning approach based on stream functions, specifically designed for AUVs. A majority of AUVs separate depth control and path control, making two-dimensional path planning more feasible than three-dimensional. Notably, Forward-Looking Sonar (FLS) sensors enable three-dimensional navigation, but they mainly scan laterally, usually providing information in the xy plane. Given these observations, the research accentuates the utility of a two-dimensional stream function. Introduced herein is a streamline function methodology for singular obstacle evasion, with mathematical validations elucidating the generated stream path's aptitude for obstacle circumvention sans local minima confrontations. To prove the effectiveness of the proposed path-generation method, the path planning was conducted in accordance with the requirements of the FLS sensor installed on the newly developed LIG Nex1 AUV. The LIG Nex1 AUV, developed by LIG NEX1, is the AUV chosen as the target system for this study. The efficiency of the designed path model was then assessed through detailed simulations in collaboration with the on-board control system.

This paper presents two contributions. First, a mathematical analysis of the streamline-based obstacle avoidance algorithm has been performed, demonstrating that the proposed technique does not encounter a local minimum. Second, the effectiveness of the path-planning algorithm has been validated through simulations based on AUV dynamics, using obstacle detection sensors on an actual AUV model.

The subsequent sections of this paper are structured as follows. Section 2 delves into preliminaries on the stream function and its path-planning applications. Section 3

explicates the proposed methodology. Section 4 presents a case study accompanied by simulation results. Finally, Section 5 concludes the paper.

## 2. Preliminaries

The study of flows of incompressible, inviscid and irrotational fluids hinges on the potential function, a fundamental concept that allows for the formal representation of such flows. The stream function, a broader mathematical construct, encapsulates both spatial coordinates and temporal progression. Its existence is intrinsically tied to the principles of continuity and incompressibility that govern fluid dynamics, with empirical observations indicating its presence even in viscous media [16].

In the framework of infinitesimal increments, denoted as $\delta x$ and $\delta y$, velocity components aligned to the respective axes, represented as $u$ and $v$, adhere to the following relation:

$$u = -\frac{\partial \psi}{\partial y}, \quad v = \frac{\partial \psi}{\partial x} \tag{1}$$

Here, the velocities $u$ and $v$ pertain to the $x$ and $y$ directions, correspondingly. The term $\nabla^2 \psi$ epitomizes the vorticity of fluid flow. An irrotational flow within a domain $\Omega$ satisfies $\nabla^2 \psi = 0$, $\psi$ is a harmonic function on $\Omega$ [17]. It is a well-known fact that harmonic functions do not possess any local optimum [18]. Consequently, stream functions also lack local optimum.

This work harnesses the representation of the stream function through complex equations. In this context, the marriage between stream and potential functions yields the following complex equation:

**Definition 1.** *Complex Potential Given $\phi$ and $\psi$ as the velocity potential and stream function describing the irrotational bidimensional movements of an inviscid fluid, the associated complex potential $\omega$ is given by:*

$$\omega = \phi + i\psi \tag{2}$$

*This results in the following velocity components:*

$$\frac{\partial \phi}{\partial x} = \frac{\partial \psi}{\partial y}, \quad \frac{\partial \phi}{\partial y} = -\frac{\partial \psi}{\partial x} \tag{3}$$

*This relationship precisely aligns with the Cauchy–Riemann equations, signifying that $\omega$ operates as a holomorphic function of the complex variable $z = x + iy$, in regions where $\phi$ and $\psi$ are singular.*

A fundamental proposition in using stream functions within path-planning algorithms stems from the inherent ability of the stream function to depict a continuous, smooth trajectory devoid of local extremes. The crux lies in devising a complex stream function sensitive to the positional nuances of obstacles, thereby facilitating an intricate obstacle avoidance technique in alignment with streamlines.

Upon introducing an obstacle into a flow, the incumbent boundary condition necessitates the flow to be tangential to the obstacle's surface. This is in tandem with the consistent nature of the stream function on an obstacle surface, as $\psi$ maintains uniformity along a streamline. To identify a streamline that envelopes the specified obstacle, one must calibrate the imaginary component of the complex potential function to be consistent.

The subsequent theorem, known as the Circle Theorem, elucidated in [16], delves into the dynamics of the complex potential with an integrated boundary condition.

**Theorem 1.** *Circle Theorem [16] considers an irrotational, bidimensional flow of an incompressible inviscid fluid across the z-plane devoid of rigid boundaries. If the complex potential of the flow is denoted by $f(z)$, and all singularities of $f(z)$ lie beyond a distance r from a reference point b, then*

the introduction of a circular cylinder, characterized by its cross-sectional circle C where $|z - b| = r$, alters the complex potential as:

$$\omega = \phi + i\psi = f(z) + \bar{f}\left(\frac{r^2}{z-b} + \bar{b}\right) \tag{4}$$

**Proof.** A detailed proof is articulated in [16]. $\square$

Here, $\bar{f}$ denotes the conjugate function, while $\bar{b}$ represents the associated conjugate variable in the complex plane. The obstacle $b$ is expressed in its complex form as $b = o_x + o_y i$, where $o_x$ and $o_y$ designate the coordinates of the obstacle in the $x$ and $y$ axes, respectively. Similarly, the position (z) is expressed as $z = x + yi$.

The Circle Theorem provides a framework for constructing the vehicle's stream function using primitives located in arbitrary positions. A particularly salient primitive for AUVs is the sink, represented by $f_s$:

$$f_s(z) = -K\ln(z) \tag{5}$$

where $K \in \mathbb{R}^+$ denotes the strength of the associated singularity.

Given that the sink is positioned at the origin and an obstacle is situated at $(o_x, o_y)$, the complex potential $\omega$, by invoking the Circle Theorem, becomes:

$$\omega = -K\ln(z) - K\ln\left(\frac{r^2}{z-b} + \bar{b}\right) \tag{6}$$

The ensuing stream function, derived from the imaginary component of Equation (6), is expressed as:

$$\psi(z) = \psi_s(z) + \psi_o(z) = -K\tan^{-1}\left(\frac{y}{x}\right) - K\tan^{-1}\left(\frac{\frac{r^2(y-o_y)}{(x-o_x)^2+(y-o_y)^2} + o_y}{\frac{r^2(x-o_x)}{(x-o_x)^2+(y-o_y)^2} + o_x}\right) \tag{7}$$

In the above relation, $\psi_s$ and $\psi_o$ denote the stream function components for the sink and the obstacle, respectively. The detailed derivation of Equation (7) is explained in Appendix A.

## 3. Obstacle Avoidance Path Based on Stream Function

### 3.1. Design of Stream Function for Path Planning

In this study, a novel path-planning approach for AUV in a two-dimensional plane is introduced. This methodology requires three pivotal coordinates: the starting point, the location of the obstacle and the destination. The stream function, as articulated in Equation (7), omits the target location, maintaining the origin as the fixed starting point. Consequently, the objective is to devise a flow function that encompasses the target location. A comparison is made between the conventional stream function and the proposed approach, in which the source primitive symbolizes the target location, and the sink primitive denotes the start position. The complex potential function (6) is extended to incorporate this additional dimension, detailed as follows:

$$\omega = -K\ln(z-s) + K\ln(z-g) - K\ln\left(\frac{r^2}{z-o} + \bar{o}\right) \tag{8}$$

The coordinates of the starting and goal points are represented by $s = s_x + s_y i$ and $g = g_x + g_y i$, respectively. Consequently, the stream function is calculated as follows:

$$\psi(z) = \psi_s(z) + \psi_g(z) + \psi_o(z)$$

$$= -K \tan^{-1}\left(\frac{y - s_y}{x - s_x}\right) + K \tan^{-1}\left(\frac{y - g_y}{x - g_x}\right) - K \tan^{-1}\left(\frac{\frac{r^2(y - o_y)}{(x - o_x)^2 + (y - o_y)^2} + o_y}{\frac{r^2(x - o_x)}{(x - o_x)^2 + (y - o_y)^2} + o_x}\right) \quad (9)$$

Here, $\psi_g(z)$ represents the stream function component for the goal point.

Given the AUV's heading as $\theta_h$, the waypoint for an avoidance path can be computed using the following relations:

$$x(t + 1) = x(t) + L \cos \theta_h(t), \quad (10)$$

$$y(t + 1) = y(t) + L \sin \theta_h(t), \quad (11)$$

where $L$ denotes the movement distance and $\theta_h(t)$ represents the heading angle at time $t$. This formulation transforms the coordinate components of the waypoint determination from $(x, y)$ to $(L, \theta_h(t))$. The core objective of this investigation is to identify the optimal heading angle, $\theta_h(t)$, ensuring that the waypoint $(x, y)$ is contained within $\mathbb{C}_f$. It is noteworthy that the movement distance $L$ is inherently determined by the vehicle's speed.

In the context of this study, the stream function plays a crucial role in determining the optimal heading angle. The objective is to ascertain the optimal heading angle, $\theta_{opt}(t)$, such that it minimizes a defined cost function. Formally, the problem can be framed as:

$$\theta_{opt}(t) = \arg \min_{\theta} f_{cost}(\theta(t)). \quad (12)$$

The function $f_{cost}(\theta(t))$ quantifies the desirability of each potential heading angle. A representative cost function, constructed using the stream function, is given by:

$$f(\theta(t)) = \|\psi(x(t) + L \cos \theta(t) + i \cdot (y(t) + L \sin \theta(t)))\|^2, \quad (13)$$

where the pair $(x(t), y(t))$ delineates the boundary coordinate of the sonar's fan-shaped search zone. Through this cost function, the magnitude of the flow function is calculated, with $\theta_{opt}(t)$ being the angle that yields the minimal value of the flow function.

To bolster the robustness of the approach, this study introduces a histogram-based technique for the cost function. Specifically, a radial histogram formulation is proposed:

$$f_{cost}(t) = \sum_{l=0}^{l_s} \sum_{\theta = \theta_h(t-1) - \theta_s}^{\theta_h(t-1) + \theta_s} \|\psi(x_0 + l \cos \theta + i \cdot (y_0 + l \sin \theta))\|^2, \quad t > 2, \quad (14)$$

where $\theta_h(t-1)$ denotes the AUV's prior heading angle. The parameter $l_s$ signifies the maximum detection range of the FLS and $\theta_h(0)$ represents the AUV's initial heading angle.

### 3.2. Characteristic of Forward Looking Sonar

A crucial challenge in AUV deployment concerns the measurement methodologies adopted. Predominantly, sonar sensors are employed for obstacle detection, with their emitted acoustic signals undergoing a series of intricate signal and image processing transformations prior to visualization. Although this manuscript does not delve extensively into the nuances of image processing or enhancements in sonar efficiency, the repercussions of these preprocessing phases warrant consideration. This discourse is predicated on several foundational assumptions:

- Every obstacle within the search area is identified post-preprocessing.
- The discerned data include only the position and distance of the obstacles.
- Pre-processing procedures are sufficiently rapid to facilitate real-time operations.

In the present investigation, the characteristics of the FLS sensor were incorporated to enhance the accuracy of the path planning for AUVs. The FLS sensor, integrated into the LIG AUV, covers a scanning ambit of 120 degrees, systematically segmented into 60 bins. Each bin, with an angular breadth of 2 degrees, furnishes data on five discrete distances. Furthermore, the vertical scanning range of the sensor extends to 15 degrees. The fundamental coordination of the FLS is depicted in Figure 1a and images captured from the actual sensor are presented in Figure 2. Within these figures, obstacles are denoted by a red dot, whereas dots of varying colors signify reflections of diminished intensity. The designated search angle in the figure is 90 degrees with a maximum detectable range of 100 m. As a result, the FLS of the LIG Nex1 AUV produces a matrix, denoted as $D_{FLS}$, described as:

$$\mathbb{D}_{FLS} = \begin{bmatrix} d_{1,1} & d_{1,2} & \dots & d_{1,5} \\ d_{2,1} & d_{2,2} & \dots & d_{1,5} \\ \dots & \dots & \dots & \dots \\ d_{60,1} & d_{60,2} & \dots & d_{60,5} \end{bmatrix} \tag{15}$$

where $d_{i,j}$ denotes the distance of the target with the $j$th intensity at the $i$th angle. If no obstacle is detected at the $i$th scanning angle, then $d_{i,jk} \forall k$ is set to 0.

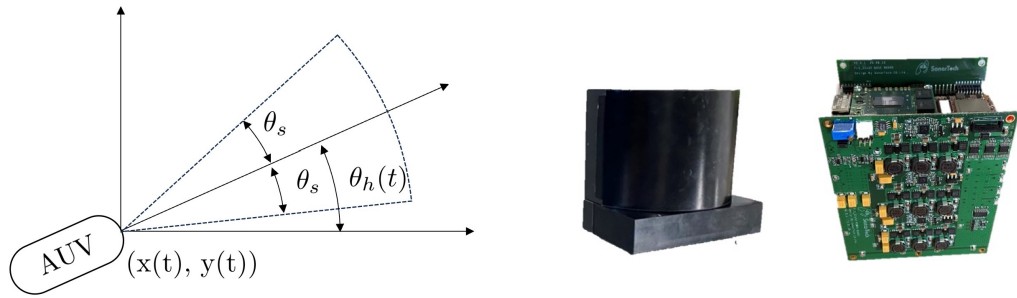

(**a**) Coordination of FLS

(**b**) Sensor and Processing board of FLS

**Figure 1.** FLS of LIG Nex1 AUV.

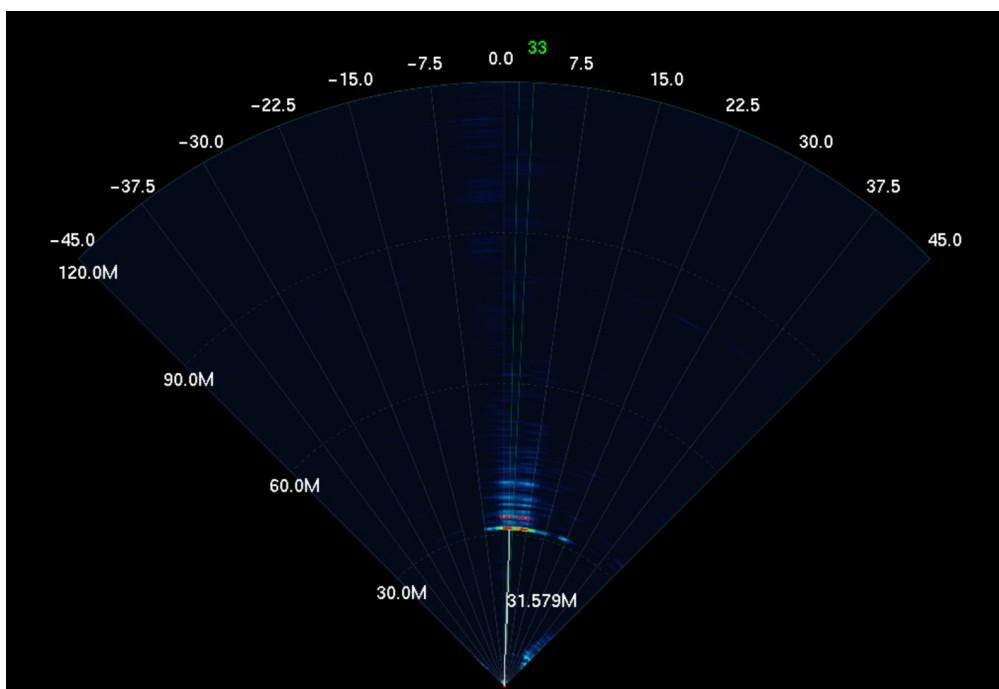

**Figure 2.** Raw data visualization of LIG Nex1 AUV's FLS sonar.

In conventional marine navigation settings, encounters with various obstacles are sporadic. Although path planning for terrestrial robots in intricate environments has been thoroughly examined, underwater terrains inherently span three dimensions. Floating objects, potentially perceived as underwater obstacles, are seldom observed in operational contexts. A particularly intricate component involves detecting obstacles and gauging their positions using sensors like the FLS. This study delves into a technique for obstacle recognition leveraging $\mathbb{D}_{FLS}$ data, seamlessly integrating this identification within the streamline path-planning approach. Let $\theta_i^B$ denote the scanning angle of the $i$th bin of the FLS sensor and let $B(\theta)$ represent the index value of the bin that encompasses an arbitrary angle $\theta$. Figure 3 illustrates an example of $\mathbb{D}_{FLS}$ generated by LIG Nex1 AUV's FLS during the detection of single obstacles. Obstacles manifest as clusters in the values, and based on the perspective of the unmanned submersible, the distances from the obstacle can be derived from the index values of the most left angle $\theta_l^B$ and the most right angle $\theta_r^B$. When an obstacle is detected, evasion must be carried out based on the information of the nearest obstacle. Thus, for each bin, the information about the closest obstacle to evasion can be defined as follows:

$$D_i = \min_{k=1}^{5} d_{i,k} \tag{16}$$

The size of the detected obstacle $o_s$ can be estimated as follows when a single obstacle is identified:

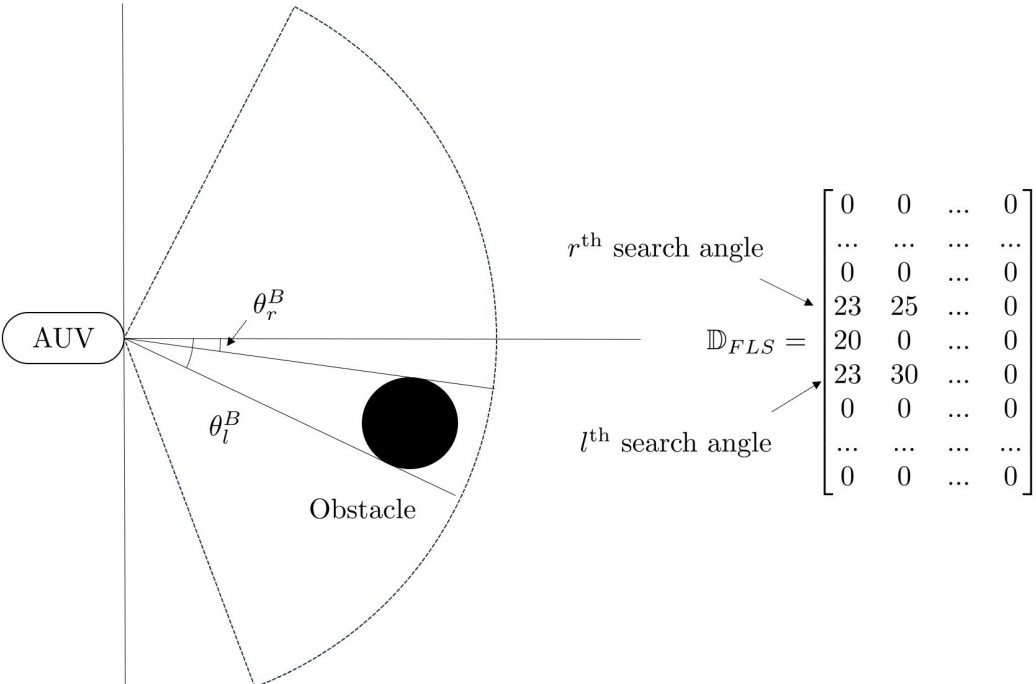

**Figure 3.** Example of the search angle and result matrix values when an obstacle is detected by the FLS.

$$o_s = \frac{\left|\theta_l^B - \theta_r^B\right|}{\left|B(\theta_r^b) - B(\theta_l^b)\right|} \sum_{\theta=\theta_l^B}^{\theta_r^B} D_\theta \tag{17}$$

Furthermore, the location of the obstacle $(o_x, o_y)$ and the distance between obstacle and AUV $o_d$ can be estimated as described below,

$$o_d = \frac{1}{\left|B(\theta_r^b) - B(\theta_l^b)\right|} \sum_{\theta=\theta_l^B}^{\theta_r^B} D_\theta \tag{18}$$

$$o_x = x + o_d(k) \cos\left(\frac{\theta_l^B + \theta_r^B}{2}\right) \tag{19}$$

$$o_y = y + o_d(k) \sin\left(\frac{\theta_l^B + \theta_r^B}{2}\right) \tag{20}$$

where $x, y$ are the position of the AUV.

### 3.3. Design of Avoidance Path Planning for Single Obstacle

In general aquatic environments, excluding the topography, most obstacles in a floating state are often singular. Therefore, the initial focus is on the methodology for generating obstacle avoidance paths in the presence of a single floating obstacle. Assume that there are $n$ waypoints generated by the global path technique. If the $k$th waypoint is denoted as $P(k) = (x_p(k), y_p(k))$, then the obstacle avoidance path generated when moving to the next waypoint $(k + 1)$th can be formed using the stream function $\psi_s(z)$ such as defined:

$$\psi_s(z) = |\psi_k(z) + \psi_{k+1}(z)| + |g(\psi_o(z))| \tag{21}$$

$$\psi_k(z) = -K \tan^{-1}\left(\frac{y(t) - y_p(k)}{x(t) - x_p(k)}\right) \tag{22}$$

$$\psi_{k+1}(z) = K \tan^{-1}\left(\frac{y(t) - y_p(k+1)}{x(t) - x_p(k+1)}\right) \tag{23}$$

where the function $g(\psi_o(z))$ is defined as:

$$g(\psi_o(z)) = \begin{cases} \psi_o(z) & \text{if } o_d \geq o_s \\ \rho_o & \text{otherwise} \end{cases} \tag{24}$$

where $\rho_o$ denotes the minimum bias value of the obstacle's stream function.

**Assumption 1.** *The AUV traverses between two waypoints, $P(k)$ and $P(k+1)$. When its position, denoted by $x(t)$ and $y(t)$, nears $P(k+1)$, the index $k$ increments. The criterion to determine if the AUV has reached the designated waypoint is when the distance between the AUV and $P(k+1)$ is less than the safety threshold $R_s$.*

**Theorem 2.** *Consider an AUV with a maximum radius $r_a$. Let an obstacle of radius $o_s$ be positioned at $(o_x, o_y) \in \mathbb{R}$. The AUVs avoid the obstacle between waypoint $P(k)$ and $P(k+1)$ if and only if:*

$$\rho_o > 2l_s\theta_s\|\psi_{max}\|^2 \tag{25}$$

$$\sqrt{(\Delta g_x)^2 + (\Delta g_y)^2} > \sqrt{(\Delta o_x)^2 + (\Delta o_y)^2} + r + r_a \tag{26}$$

*where $\Delta o_x = o_x - x_p(k)$, $\Delta o_y = o_y - y_p(k)$, $\Delta g_x = x_p(k+1) - x_p(k)$ and $\Delta g_y = y_p(k+1) - y_p(k)$. $\psi_{max}$ is the peak value of the stream function $\psi(z)$ outside the obstacle zone. $l_s$ is the maximum range of FLS sonar. $\theta_s$ is the maximum search angle of FLS. Then, collision-free local waypoints between P(k) and P(k+1) are calculated by:*

$$x(t+1) = x(t) + L\cos(\theta_h(t)) \tag{27}$$

$$y(t+1) = y(t) + L\sin(\theta_h(t)) \tag{28}$$

*where the optimal avoidance angle $\theta_h$ is determined as:*

$$\theta_h(t) = \arg\min_{\theta} \left[ \sum_{l=1}^{l_s} \sum_{\theta=\theta_h(t-1)-\theta_s}^{\theta_h(t-1)+\theta_s} \|\psi_s(x(t) + l\cos(\theta) + i(y(t) + l\sin(\theta)))\|^2 \right] \qquad (29)$$

**Proof.** The proof is to substantiate two propositions:

**1. Collision avoidance:** The path does not intersect the obstacle.

Given the cost function $f_{cost}$ defined for $(x, y) \notin \mathbb{O}$, the following relationship is established:

$$\sum_{\theta=\theta_h(t-1)-\theta_s}^{\theta_h(t-1)+\theta_s} \|\psi_m(x(t) + L\cos(\theta) + i(y(t) + L\sin(\theta)))\|^2 <$$

$$\sum_{\theta=\theta_h(t-1)-\theta_s}^{\theta_h(t-1)+\theta_s} \|\psi_{max}\|^2 < 2l_s\theta_s\|\psi_{max}\|^2 \qquad (30)$$

If $\rho_o > 2l_s\theta_s\|\psi_{max}\|^2$, then:

$$f_{cost}(z_1) < f_{cost}(z_2), \quad \text{where} \quad z_1 \notin \mathbb{O} \quad \text{and} \quad z_2 \in \mathbb{O} \qquad (31)$$

This ensures the absence of points $(x(t) + \cos(\theta_h), y(t) + \sin(\theta_h))$ in the obstacle region, confirming our first assertion.

**2. Path Validity:** The path begins at $P(k)$ and concludes at $P(k+1)$.

Consider two stream functions as defined by:

$$\psi_k(z) + \psi_{k+1}(z) = -C\tan^{-1}(m_1) + C\tan^{-1}(m_2) \qquad (32)$$

$$\text{where} \quad m_1 = \frac{y(t) - y_p(k)}{x(t) - x_p(k)}, \quad m_2 = \frac{y(t) - y_p(k+1)}{x(t) - x_p(k+1)} \qquad (33)$$

Similarly, the obstacle stream function can be expressed as:

$$\psi_o(z) = \tan^{-1}\left( \hat{r}\frac{y(t) + (1/\hat{r} - 1)o_y}{x(t) + (1/\hat{r} - 1)o_x} \right) \qquad (34)$$

where $\hat{r} = o_s^2 / \{(x(t) - o_x)^2 + (y(t) - o_y)^2\}$. The modified stream function is:

$$\psi_m(z) = \left| -C\tan^{-1}(m_1) + C\tan^{-1}(m_2) \right| + \left| g\left(\tan^{-1}\left( \hat{r}\frac{y(t) + (1/\hat{r} - 1)o_y}{x(t) + (1/\hat{r} - 1)o_x} \right)\right) \right| \qquad (35)$$

The optimal path is achieved by selecting the angle yielding the minimum mean square sum from Equation (41). This minimum is primarily influenced by the values of $\psi_k(z) + \psi_{k+1}(z)$, which is least when:

$$C\tan^{-1}(m_1) = C\tan^{-1}(m_2) \qquad (36)$$

This results in $\psi_k(z) + \psi_{k+1}(z)$ having a minimum at $(x, y)$ which satisfies:

$$y = \frac{y_p(k+1) - y_p(k)}{x_p(k+1) - x_p(k)}x + y_p(k) - \frac{y_p(k+1) - y_p(k)}{x_p(k+1) - x_p(k)}x_p(k) \qquad (37)$$

This equation encompasses both the starting and ending points. Additionally, all streamlines converge to $(x_p(k), y_p(k))$ and $(o_x, o_y)$ since $\psi_k(z)$ and $\psi_{k+1}(z)$ represent the primary sources of sink and source. This validates our second assertion. $\square$

### 3.4. Path Generation Considering the Motion of AUV

The torpedo-like AUV has its propellers and control fins at the after body. This strategic placement gives the AUV the ability to maneuver while maintaining forward propulsion. The control fins of an AUV largely determine the shape of its trajectory, particularly its curved path. When in a steady operational state, this force helps identify the minimum possible turning radius $R_t(u)$. Therefore, when making a plan for the trajectory, it is essential to take into account this minimum turning radius.

In a stabilized state, the external force $N_{ext}$ that influences the turning angle $\psi$ can be summarized as follows [19]:

$$\sum N_{ext} = N_{hydrostatic} + N_{lift} + N_{drag} + N_{control} \tag{38}$$

$$N_{control} = N_{uuq\delta_r} u^2 \delta_r \tag{39}$$

where $N_{hydrostatic}$ is hydrostatic force, $N_{lift}$ is body lift force and $N_{drag}$ is drag force. $\delta_r$ denotes the angle of the control fin of the rudder. Although most external forces change in relation to velocity, the control force varies in proportion to the square of the velocity. Therefore, in a stabilized state, it is possible to estimate the maximum turning angle $\psi_m(u)$ based on velocity. Estimation of the maximum turning angle $\psi_m(u)$ is commonly achieved through simulations of sea elevation. Due to variations in the hydrodynamic coefficients based on the shape of the AUV, it is challenging to define a generalized maximum value. Once the maximum turning angle is defined, it can be incorporated into the previously proposed path-planning technique.

Figure 4 presents the results of the simulation of the maximum rotation at 3 kt from the stabilized motion model mentioned above. While a driving control command for rotation was given at 6 degr/s, it can be observed that the maximum rotational speed is limited to 5 deg/s. The maximum deflection angle of the rudder is limited to 14 degrees and maintains this maximum angle during rotation, as shown in Figure 4. The maximum rotation angle was calculated based on the maximum rotation angular velocity $r_{max}$ obtained from the simulation, utilizing the FLS's maximum detection distance $l_s$ and speed $u$ as follows:

$$\psi_m(u) = r_{max} \frac{l_s}{u} \tag{40}$$

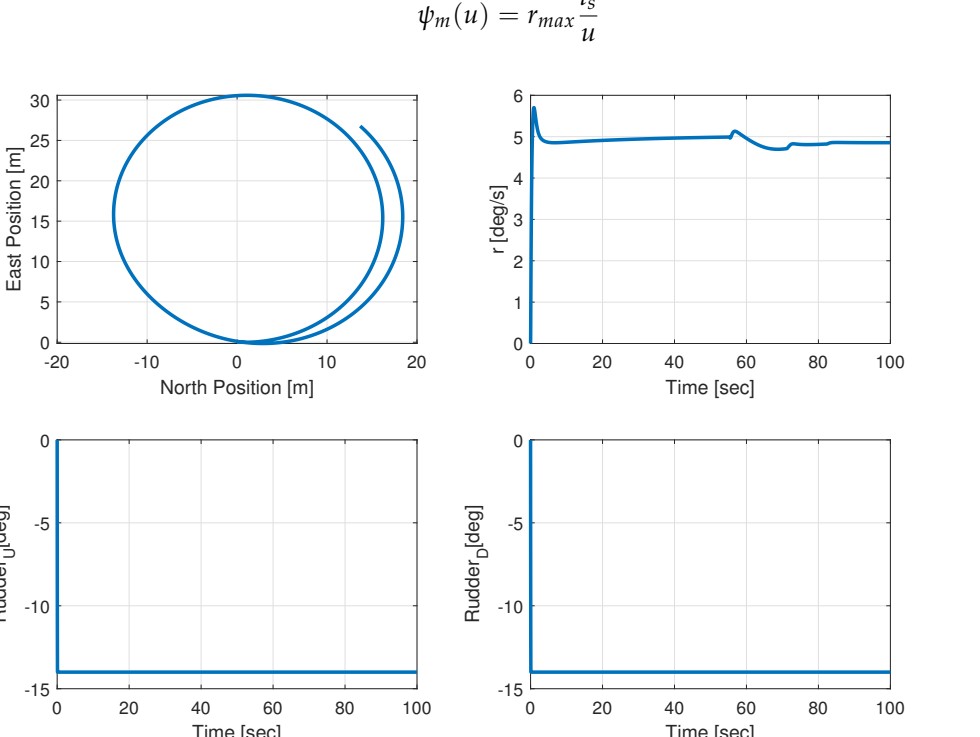

**Figure 4.** Maximum rotation radius simulation at 3 kts with 6 degree heading angle.

The proposed path-planning method is formulated around finding the optimal avoidance angle based on obstacle locations determined via FLS and the current position of the AUV. By integrating the turning constraints of the AUV into the optimal obstacle avoidance angle, a trajectory that takes into account the AUV's motion can be formulated. Then, the optimal avoidance angle $\theta_h$ considering AUV's motion constraints is calculated as:

$$\theta_h(t) = \arg\min_{\theta} \left[ \sum_{l=1}^{l_s} \sum_{\theta=\theta_h(t-1)-\theta_d}^{\theta_h(t-1)+\theta_d} \|\psi_s(x(t) + l\cos(\theta) + \mathrm{i}(y(t) + l\sin(\theta)))\|^2 \right] \quad (41)$$

where $\theta_d$ is the minimum value between the maximum turning angle $\psi_m(u)$ and the maximum detection angle $\theta_s$.

## 4. Simulation Results and Analysis

The proposed streamline-based path-planning method addresses both the generation of obstacle avoidance trajectories in intricate maritime scenarios and the facilitation of local path planning amidst waypoints. An inherent attribute of the trajectories produced through this approach is their alignment with rational principles, facilitating the navigation of AUV. The effectiveness of this method was evaluated through obstacle avoidance simulations based on the AUV motion model developed by LIG Nex1. As shown in Figure 5, the characteristics of the obstacle avoidance sensors incorporated into the LIG Nex1 AUV were considered. For an authentic simulation environment, real-world representative obstacle data derived from these sensors were utilized. The chief specification of the in-use FLS is as follows:

- Feasible operating range: 30 m.
- Horizontal beam width: $120°$.

Note that the insonified region of FLS is not included in the simulation results presented.

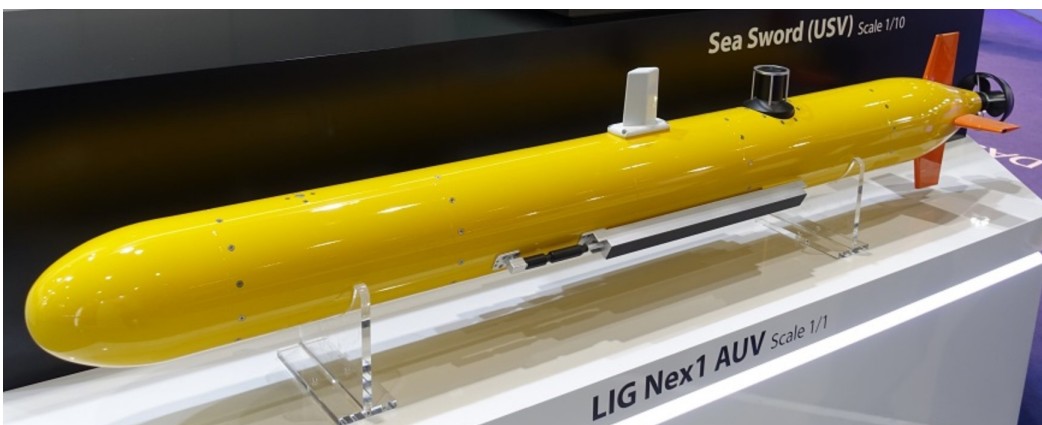

**Figure 5.** Mock-up of LIG Nex1 AUV (FLS sonar is installed in front head in real AUV).

### 4.1. Obstacle Avoidance Standalone Simulation

A comprehensive performance assessment of the path-generation algorithm was performed through simulations. These simulations were designed to both evaluate the algorithm's efficacy against generic obstacle patterns and test its robustness in overcoming obstacles leading to local minima. The primary objective is to empirically validate its robustness and feasibility. For consistency, each simulation was conducted under the assumption that the dimensions of the underwater environment are (100 m × 100 m). Furthermore, the AUV is abstracted as a particle with a radius of 2 m. The threshold bias value, $\rho_0$, is designated as $10^5$. This value sufficiently satisfies the condition given by Equation (25) in Theorem 2, particularly as $\psi_{max}$ remains below $10^3$ across all underwater environmental points.

Figure 6 highlights the local minimum dilemma often encountered in potential field methods. Here, the AUV speed is set to (1 m/s) with a sonar operating range of 20 m. Traditional potential-field techniques tend to cause the AUV to stall in front of U-shaped obstacles. On the contrary, our proposed stream function-centric approach successfully navigates these challenges, generating smooth paths for obstacle avoidance.

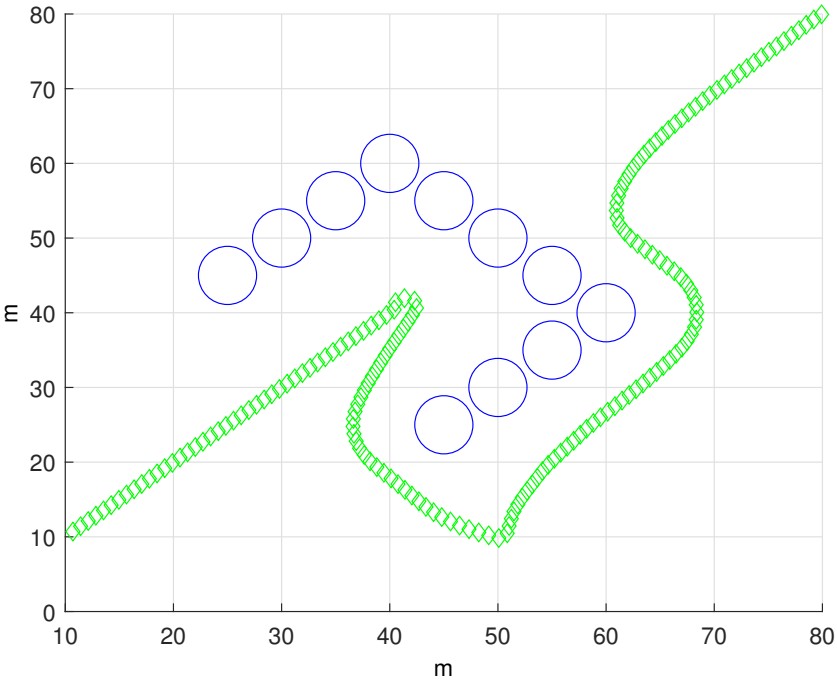

**Figure 6.** Illustration of the local minimum challenge in potential fields.

Figures 7 and 8 represent scenarios with increased navigational complexity. The algorithm consistently produces reliable avoidance paths in the majority of these complicated cases. Ensuring an AUV's precision in following the designated trajectory is paramount.

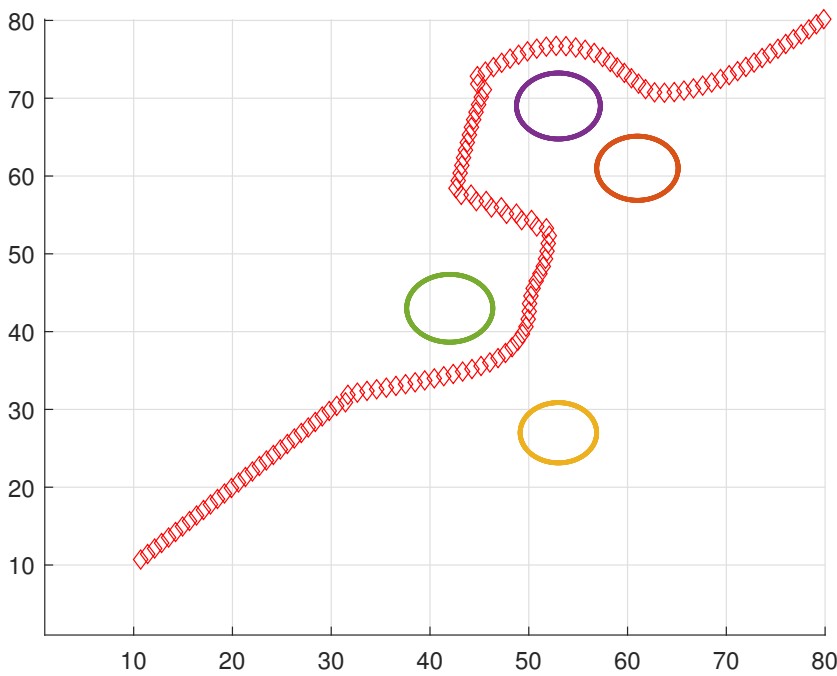

**Figure 7.** AUV avoidance path in a multifaceted environment (Scenario I).

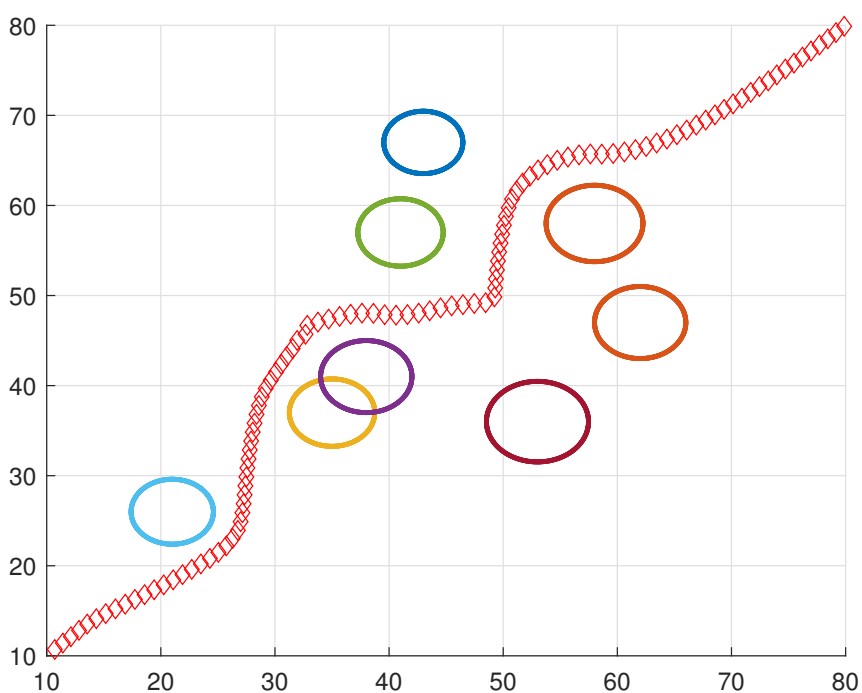

**Figure 8.** AUV avoidance path in a multifaceted environment (Scenario II).

### 4.2. Integrated Obstacle Avoidance Simulation

To determine whether the proposed route is easily navigable for the autonomous underwater vehicle, a simulation that integrates both the control algorithm and the motion model is essential. To implement the proposed obstacle generation technique in the actual control algorithm, a simulation framework was designed, as shown in Figure 9, which encompasses the AUV's motion dynamics, path-planning algorithm and virtual obstacle management. The motion dynamics, calculated in real-time, utilize a 6-degrees-of-freedom equation to derive the AUV's posture and position. The Mission Management module orchestrates AUV operations and formulates the overarching mission trajectory. Within this simulated environment, the dynamics and obstacle data converge to generate virtual obstacles, currently focusing solely on circular-shaped challenges. The obstacle avoidance sonar model, mirroring the functionalities of the FLS, identifies these virtual impediments and provides pertinent data. Core path planning, capitalizing on a streamline-based algorithm, discerns optimal traversal paths and depths. The entire navigation process culminates with a nonlinear controller guiding the AUV based on this predetermined trajectory and depth. All components were conceptualized and instantiated using MATLAB.

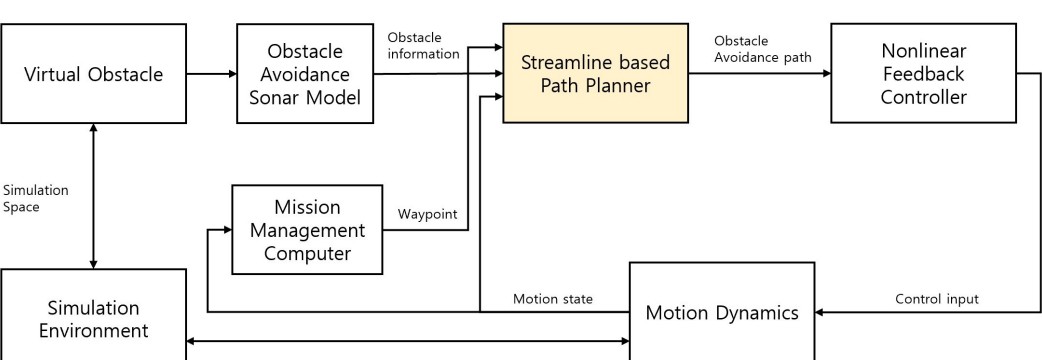

**Figure 9.** Simulation configuration for LIG AUV model.

Figure 10 illustrates the trajectories of an AUV travelling at 4 knots and navigating between waypoints 150 m apart, with a sonar detecting range of 30 m and a maximum

turning angle of 120°. The strategic placement of obstacles modulated the trajectories and the changes were highlighted. Additionally, Figure 11 shows the control output fluctuations due to depth changes for avoiding obstacles. Complicated paths may make control angles to their limits, possibly slowing down the AUV and increasing energy usage. As more obstacles are added, the navigation path lengthens, causing the control panel's operation to slow down. The chattering phenomenon in the control angle is not due to the generated path but is inherent to the nonlinear controller's characteristics. During actual navigation, many sensors provide directional input. To robustly handle this input, the controller has an uncontrolled region set, which leads to the chattering effect. The occurrence of chatter indicates a small error between the commanded and actual path, suggesting that the path is easily navigable for the autonomous underwater vehicle. Upon the second obstacle avoidance, one can observe a pronounced curve in the path as illustrated in Figure 10. However, examining the rudder angle shows that while it briefly maintains its maximum value for the initial rotation maneuver, it does not utilize the entire deflection angle for the subsequent path control. This indicates that the proposed path is traceable. The reason for maintaining the initial maximum deflection is to alter the initial direction, a phenomenon executed to avoid rapid obstacles.

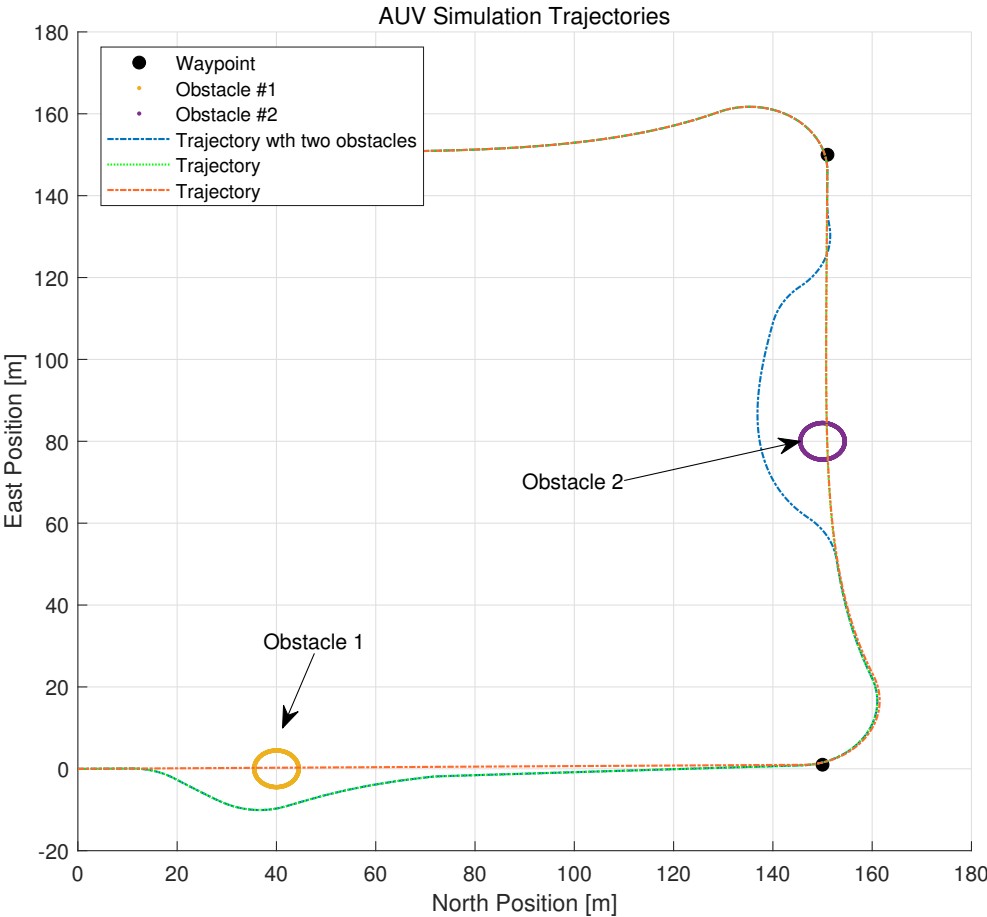

**Figure 10.** Comparison of simulation trajectories with various obstacle conditions.

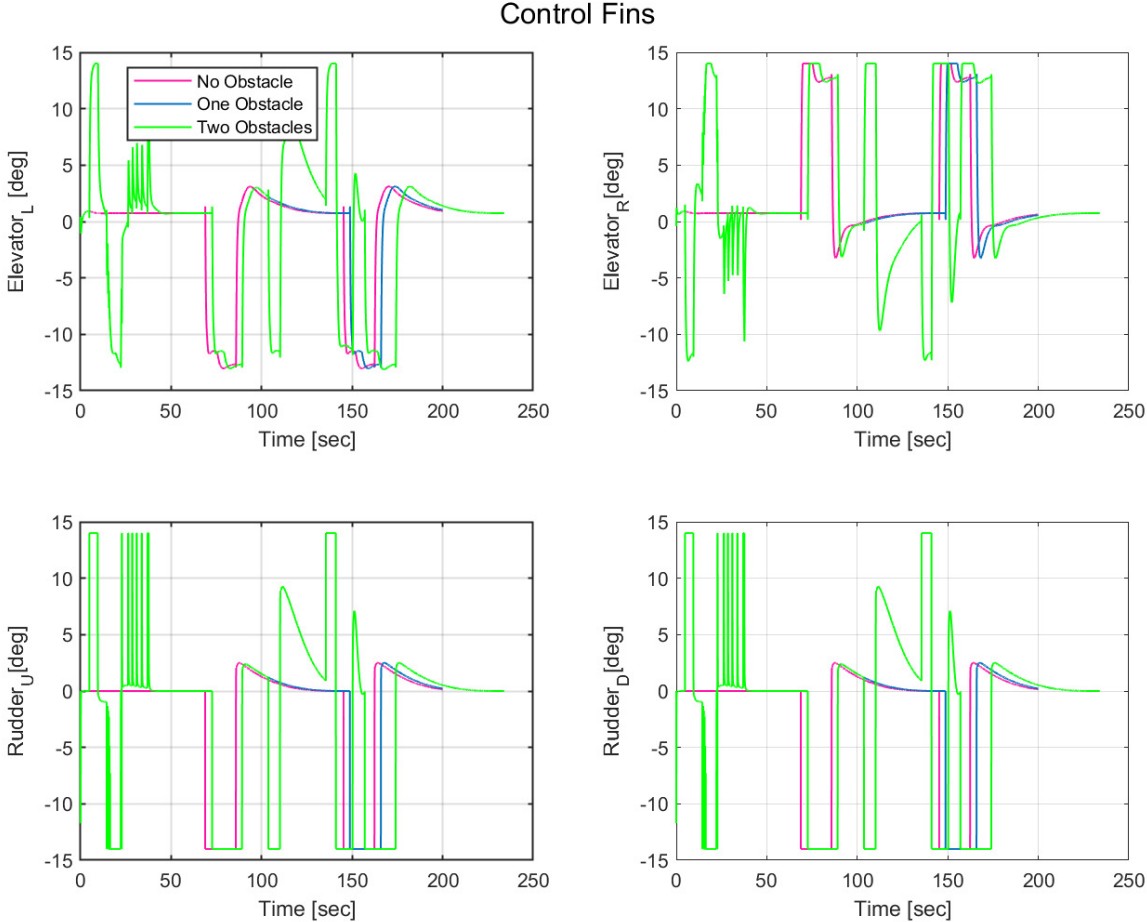

**Figure 11.** Comparison of control fins with various obstacle conditions.

## 5. Conclusions

This paper proposed the development of an obstacle avoidance algorithm for AUVs that utilizes a stream function. Recent studies have shown a growing interest in stream function-based approaches, and this work defines a path-planning problem based on the stream function to identify the optimal traversal path. It was shown that the derived optimal path for a single obstacle is free of collisions and local minima. Additionally, a methodology was proposed to generate paths within avoidance capabilities, taking into account the turning radius of torpedo-shaped AUVs. The proposed path-planning technique was then integrated with the FLS specifications used in a prototype AUV and linked to a motion model-based controller. This approach was not only used for path planning but was also extended to simulations of the entire mission, thus demonstrating its effectiveness. Future research will aim to confirm the stable obstacle avoidance in real-sea conditions through field tests, integrating the obstacle sensor and the controller in actual scenarios.

**Author Contributions:** Conceptualization, M.H.K.; methodology, M.H.K.; software, T.Y.; validation, T.Y. and M.H.K.; formal analysis, M.H.K.; investigation, M.H.K. and T.Y.; resources, S.J.P.; data curation, T.Y.; writing—original draft preparation, M.H.K.; writing—review and editing, K.O.; supervision, K.O.; project administration, M.H.K. and K.O. All authors have read and agreed to the published version of the manuscript.

**Funding:** This research received no external funding.

**Data Availability Statement:** Not applicable.

**Conflicts of Interest:** The authors declare no conflict of interest.

## Appendix A. Derivation of the Stream Function

The Circle Theorem yields the complex potential $\omega$ as:

$$\omega = -K\ln(z) - K\ln\left(\frac{r^2}{z-b} + \bar{b}\right), \tag{A1}$$

where $z$ is the complex position of a point in the flow. $b = o_x + io_y$ is the complex position of the obstacle. $\bar{b}$ is its complex conjugate. $r$ denotes the radius associated with the obstacle.

The stream function $\psi(z)$ is derived from the imaginary part of $\omega$. Leveraging the characteristic of the imaginary segment of a complex logarithm, the following is established:

$$\Im(\ln(z)) = \tan^{-1}\left(\frac{\Im(z)}{\Re(z)}\right). \tag{A2}$$

Dissecting the terms individually:
For the sink term:

$$\Im(-K\ln(z)) = -K\tan^{-1}\left(\frac{\Im(z)}{\Re(z)}\right), \tag{A3}$$

$$\text{Given } z = x + iy, \tag{A4}$$

$$\psi_s(z) = -K\tan^{-1}\left(\frac{y}{x}\right). \tag{A5}$$

For the obstacle-induced term:

$$\Im\left(-K\ln\left(\frac{r^2}{z-b} + \bar{b}\right)\right) \tag{A6}$$

Given:

$$z = x + iy,$$
$$b = o_x + io_y,$$

it is deduced as:

$$z - b = (x - o_x) + i(y - o_y), \tag{A7}$$

$$|z - b| = \sqrt{(x - o_x)^2 + (y - o_y)^2}. \tag{A8}$$

Continuing the derivation:

$$\frac{r^2}{z-b} = \frac{r^2(x - o_x)}{(x - o_x)^2 + (y - o_y)^2} + i\frac{r^2(y - o_y)}{(x - o_x)^2 + (y - o_y)^2}. \tag{A9}$$

Combining with $\bar{b}$, the real and imaginary components are:

$$\Re\left(\frac{r^2}{z-b} + \bar{b}\right) = \frac{r^2(x - o_x)}{(x - o_x)^2 + (y - o_y)^2} + o_x, \tag{A10}$$

$$\Im\left(\frac{r^2}{z-b} + \bar{b}\right) = \frac{r^2(y - o_y)}{(x - o_x)^2 + (y - o_y)^2} + o_y. \tag{A11}$$

Extracting the imaginary component:

$$\psi_o(z) = -K\tan^{-1}\left(\frac{\frac{r^2(y-o_y)}{(x-o_x)^2+(y-o_y)^2} + o_y}{\frac{r^2(x-o_x)}{(x-o_x)^2+(y-o_y)^2} + o_x}\right). \tag{A12}$$



Incorporating both the sink and obstacle components, the total stream function becomes:

$$\psi(z) = \psi_s(z) + \psi_o(z). \tag{A13}$$

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
