# Peer review of "Forward-Looking Sonar-Based Stream Function Algorithm for Obstacle Avoidance in Autonomous Underwater Vehicles"

_jmse, doi:10.3390/jmse11101998_

Round 1

Reviewer 1 Report

The authors propose an improvement to a stream function path planning claiming that it guarantees not to get stuck in a local minimum.

Abstract: The following is a tautology (or even worse): The proposed algorithm is based on a stream function that is derived from the stream function."

Line 118: Not clear what does "en" mean. "be"?

Line 164: What does the last hieroglyph signify?

Figure 9. Not clear why simulated AUV paths differ (for obastacle 1) in cases of one and two obstactles. While avoiding obstacle 1 AUV cannot know about the existence of obstacle 2.

Abstreact: The following is a tautology (or even worse): The proposed algorithm is based on a stream function that is derived from the stream function."

Line 118: Not clear what does "en" mean. "be"?

Line 164: What does the last hieroglyph signify?

Figure 9. Not clear why simulated AUV paths differ (for obastacle 1) in cases of one and two obstactles. While avoiding obstacle 1 AUV cannot know about the existence of obstacle 2.

Author Response

Thank you for your constructive feedback on our manuscript. We have made diligent efforts to address and rectify the issues you highlighted. The comments you pointed out have been addressed and compiled in the attached document. Following the feedback from Reviewer 1, we were able to identify and rectify a previously overlooked program error in the final simulation. We deeply appreciate the valuable insights provided by Reviewer 1.

Reviewer 2 Report

The article considers the problem of AUV path planning which is very relevant nowadays due to extensive development of underwater vehicles and their increasing role in different applications. Authors propose using stream functions for obstacle avoidance based on sonar data. The work surely presents a scientific novelty, however there are some issues which must be fixed:

1) The biggest issue for me is the sloppiness of presented mathematics and research in general. For example, denotions from all the equations in the first part of the article are largely not explained: What is psi in (2.1)? What are r and b in (2.5) ? Should not be there w(z) also? What is variable d in line 120? Explaining of all these denotions could significantly improve the text in my opinion.

2) The statement of the problem is rather unclear. Where are variables l and l_s in sum in Equation (3.7) ? Theorem 3.2 repeats this error in Equation (3.22). Moreover, now there is min instead of argmin. Please fix all this mess. 

3) More importantly, where are equations of dynamic model of the AUV? Is its motion constrained by its turning radius, which authors mention for the first time on Page 8 in Theorem 3.2 ? Does it have inertia due to its mass and surrounding water? Can it turn in one place? Where is the guarantee that obtained avoidance angle can be achieved by the system at right time at all? Please elaborate on this.

4) Mathematical signs "=", "+" and so on must be placed on both lines before and after line break in all Equations. Please fix this in (2.6), (3.2) and etc.

5) Line 106 mentions Circle Theorem and [19] reference, however [19] does not contain any such theorem, so please fix the reference or the words around it.

6) How is the choice of function f_s from Equation (2.4) motivated? There is a lack of explanation if the text for this crucial formula.

7) How is (2.6) 'derived'? This is unclear and some citation should be placed there at least.

8) The references are out of order in Introduction. They must appear according to their numbering, but the first one mentioned is [18].

9) Section 4 does not contain any comparison of presented method with other, for example with traditional ones based on virtual potential functions. What is the quantative advantage of the algorithm in question?

After fixing all the issues mentioned above, the article can be considered for publication in my opinion.

I think that moderate English editing is required in general and all the errors and misspellings should be fixed, for example:

1) Abstract, line 6: "The proposed algorithm is based on a stream function that is derived from the stream function." – That is nonsense. 

2) Line 14: "AUV are becoming..." – AUVs

3) Line 108: "Proffers" - ???

4) Line 118: "can en calculated" – ???

and so on.

Author Response

Thank you for your constructive feedback on our manuscript. We have made diligent efforts to address and rectify the issues you highlighted. The comments you pointed out have been addressed and compiled in the attached document. With the assistance of Reviewer 2, significant improvements were made to address the mathematical errors and ambiguities in the formula development of the submitted manuscript. Deep appreciation is extended to Reviewer 2 for the meticulous examination of both the mathematical details and content.

Reviewer 3 Report

1.       Underwater navigation including obstacle avoidance especially by sonar is one of the leading scientific problems of recent years. Both in civilian and military applications. In this paper forward looking sonar was proposed to AUV anticollision solution.

2.       General remarks

a.       Too many abbreviations make it difficult to follow the content of the article. Each abbreviation should be expanded the first time it appears. Not all readers need to know all abbreviations. Also, in the title of pictures (for example Fig.1).  LIG AUV should be introduced before using this name before using in line 68. Abbreviation AUV is well known but also should be introduced before first use in the article earlier then in line 114. Where is described LIG NEX1 AUV?

b.       Titles of drawings should be described in one sentence and the description of the content of the drawing in the text preceding the drawing – Fig.2.

c.       Please use the language of a scientific research report without personal references: like “we” or “our”.

d.       The subject of underwater target detection with sonar is introduced in many papers. It is worth to compare algorithms used in acoustic source analysis idea to the solution proposed in the paper. Publications worth analyzing: doi: 10.1515/pomr-2017-0004 and doi: 10.3390/rs13051014.

e.       However the article is very well written should be carefully edited. Only few remarks included below.

3.       Specific remarks

a.       Why in formula 3.9 is k=1 to 5?

b.       What means “1)” in formula 3.11?

Author Response

Thank you for your constructive feedback on our manuscript. We have made diligent efforts to address and rectify the issues you highlighted. The comments you pointed out have been addressed and compiled in the attached document. The valuable feedback on the paper's English expressions and basic specifications has greatly improved the quality of the manuscript. We deeply appreciate Reviewer 3's thorough review and effort.

Round 2

Reviewer 2 Report

Authors have done a huge work in a very short time. The article has been extensively rewritten and has become much clearer, almost all of my questions have been answered and explained in a cover letter. I believe now it can be published, but please fix the citations numbering in Introduction, that is still out of order.

Minor editing can be done, but generaly the text is okay.

Reviewer 3 Report

My sugestions are fulffiled. Article could be finally processed.